# Molecular Variability of the *Fusarium solani* Species Complex Associated with Fusarium Wilt of Melon in Iran

**DOI:** 10.3390/jof9040486

**Published:** 2023-04-18

**Authors:** Fatemeh Sabahi, Zia Banihashemi, Maryam Mirtalebi, Martijn Rep, Santa Olga Cacciola

**Affiliations:** 1Department of Plant Protection, College of Agriculture, Shiraz University, Shiraz 7144165186, Iran; 2Molecular Plant Pathology, University of Amsterdam, 1098 XH Amsterdam, The Netherlands; 3Department of Agriculture, Food and Environment (Di3A), University of Catania, 95123 Catania, Italy

**Keywords:** FSSC, *Neocosmospora falciformis*, *Neocosmospora keratoplastica*, *Neocosmospora pisi*, *Neocosmospoa* sp., *Cucumis melo*, pathogenicity, phylogenetic analysis, haplotypes

## Abstract

Species of the *Fusarium solani* species complex (FSSC) are responsible for the Fusarium wilt disease of melon (*Cucumis melo*), a major disease of this crop in Iran. According to a recent taxonomic revision of *Fusarium* based primarily on multilocus phylogenetic analysis, *Neocosmospora*, a genus distinct from *Fusarium sensu stricto*, has been proposed to accommodate the FSSC. This study characterized 25 representative FSSC isolates from melon collected in 2009–2011 during a field survey carried out in five provinces of Iran. Pathogenicity assays showed the isolates were pathogenic on different varieties of melon and other cucurbits, including cucumber, watermelon, zucchini, pumpkin, and bottle gourd. Based on morphological characteristics and phylogenetic analysis of three genetic regions, including nrDNA internal transcribed spacer (ITS), 28S nrDNA large subunit (LSU) and translation elongation factor 1-alpha (*tef1*), *Neocosmospora falciformis* (syn. *F*. *falciforme*), *N*. *keratoplastica* (syn. *F*. *keratoplasticum*), *N. pisi* (syn. *F*. *vanettenii*), and *Neocosmospora* sp. were identified among the Iranian FSSC isolates. The *N. falciformis* isolates were the most numerous. This is the first report of *N. pisi* causing wilt and root rot disease in melon. Iranian FSSC isolates from different regions in the country shared the same multilocus haplotypes suggesting a long-distance dispersal of FSSC, probably through seeds.

## 1. Introduction

Melon (*Cucumis melo* L.) is one of the most economically important horticultural crops among the *Cucurbitaceae* and comprises diverse varieties, such as *C. melo* L. var. *cantalopensis* Naudin (cantaloupe) and *C. melo* L. var. *indorus* Naudin (long melon). The top 10 melon producing countries are China, Turkey, India, Kazakhstan, and Iran in Asia, Egypt in Africa, Spain in Europe, United States, Guatemala, and Mexico in America [1]. Iran, where melon cultivation probably dates back more than 5000 years [2], is the fifth producing country in the world. The oldest archaeological finds of melon crop are from China and Iran and are seeds dating back to 3000 B.C. [3]. Consequently, Iran was proposed as a putative center of domestication of melon [4,5]. Like other crops, melon is affected by several fungal and oomycete diseases that reduce yield and fruit quality [6]. *Fusarium oxysporum* f. sp. *melonis* [7,8,9,10], *Phytophthora melonis* [11], *Monosporascus cannonballus* [12], *Paramyrotheium foliicola* [13], *Plectosphaerella cucumerina* [14], and *Neoscytalidium hyalinum* [15] are among the most important fungal pathogens reported from melons in Iran.

The *Fusarium solani* species complex (FSSC) comprises filamentous fungi with a worldwide distribution, which causes disease in many economically important crops [16,17]. As a whole, FSSC has a wide host range of over 100 agricultural crops and typically causes foot and root rot in host plants [16,17,18]. The infected plants show wilting, stunting, chlorosis, and stem lesions [17]. This disease is known as Fusarium wilt or Fusarium root rot. The name Fusarium wilt is also used to indicate a vascular disease caused by members of the *F. oxysporum* species complex [19]. In addition to being plant pathogens, members of FSSC comprise an important group of clinical filamentous fungi, causing infections in humans and animals [20,21,22,23]. Some of the FSSC members, which are known as opportunistic human or animal pathogens, are also plant pathogens [24]. 

Molecular phylogenetic analyses of internal transcribed spacer (ITS), 28S ribosomal DNA, and *TEF-1α* gene sequences revealed that the FSSC is highly variable and comprises at least 60 phylogenetic species in three distinct subgroups, clades 1, 2, and 3 [21,22,25,26,27,28]. Clade 1 includes only two known species (*F. illucidens* and *F. plagianthi*) from New Zealand. Clade 2, as outlined by O’Donnell [27], includes the soybean sudden death syndrome pathogen [29]. Clade 3, which includes at least 35 phylogenetic species [22], is the most numerous and common group associated with plant diseases and human infections [21]. The most common species in this clade are FSSC 1 (*F. petroliphilum*), FSSC 2 (*F. keratoplasticum*), FSSC 3 + 4 (*F. falciforme*), FSSC 5 (*F. solani sensu stricto*), and FSSC 6 (*F. metavorans* sp. nov.) [28,30,31,32,33]. After the introduction of new nomenclatural rules and multilocus phylogenetic analysis as a major taxonomic criterion, both the definition and the concept of *Fusarium* as a genus have evolved [34,35]. In particular, *Neocosmospora*, accommodating the FSSC, has been proposed as a genus distinct from *Fusarium sensu stricto* (*s.s.*), and the nomenclature of species within this complex has been adapted accordingly [36,37]. Although the segregation of *Neocosmospora* as a separate genus from *Fusarium s.s.* has been questioned and is still controversial [38,39], the generic name *Neocosmospora* for members of the FSSC is gradually entering common use. For instance, it has been used to indicate the fusarioid species associated to Fusarium dry root rot of citrus in South Africa and to Fusarium rot of *Cactaceae* and other succulent plants in Iran [40,41]. 

In Iran, FSSC species have been reported as causal agents of diseases in cucurbits. For instance, *F. solani* f. sp. *cucurbitae* (Fsc) race 1 was first reported as a causal agent of root, crown, and fruit rot in cucurbits from Khorasan Razavi, Northern Khorasan, and Fars provinces [42]. In 2011, several isolates under the name of *F. solani* were reported from cucurbits in Kermanshah, Mazandaran, Fars, Khorasan, and Semnan provinces [43,44]. So far, identification of FSSC isolates from cucurbits in Iran has mostly relied on morphological and pathogenic characteristics of the causal agent; although molecular markers such as restriction fragment length polymorphism (RFLP), random amplified polymorphic DNAs (RAPDs), and fluorescent amplified fragment length polymorphism (FAFLP) were also used to explore the genetic diversity of Fsc race 1 [45,46]. Hence, detailed information on the geographic distribution, host range, and phylogenetic position of FSSC isolates recovered from melons in Iran is mostly missing. Considering the economic importance of melon cultivation in Iran, a better molecular characterization of the FSSC as a causal agent of wilt and root rot is warranted. The objectives of the present study were to (i) obtain FSSC isolates from melon plants with symptoms of Fusarium wilt sampled in major melon growing regions in Iran, (ii) assess the pathogenicity and host range of these FSSC isolates on a set of species of the *Cucurbitaceae* family, and (iii) analyze phylogenetically and identify these isolates.

## 2. Materials and Methods

### 2.1. Survey, Sampling, and Fungal Isolation

From 2009 to 2011, a field survey was performed across melon growing regions of Central (Isfahan and Yazd), Eastern (Khorasan), Northern (Semnan), and Southern (Fars) provinces in Iran. Muskmelon plants showing wilting, crown, and root rot symptoms were sampled and brought to the laboratory for further analyses. Lower stems and upper roots of both severely decayed and wilted plants were cut into 0.5–1 cm segments, surface-sterilized by dipping into 1% sodium hypochlorite for 2 min, rinsed three times in sterile distilled water (SDW), and placed in Petri dishes onto acidified potato dextrose agar (PDA; potato extract 300 g/L, dextrose 20 g/L, agar 15 g/L). Dishes were incubated at 25 °C for 3–5 days in the dark, and the resulting colonies were purified by single conidium isolation. All recovered *Fusarium* isolates were preserved by both the sterile soil and cellulose filter paper methods [47] for further use and deposited in the fungal collection of the Plant Protection Department (Shiraz University, Iran). Furthermore, a FSSC isolate obtained from J. Armengol, Universidad Politécnica de Valencia, Spain, was included as a reference in the pathogenicity assays and molecular phylogenetic analysis.

### 2.2. Morphological Characterization

In order to study the in vitro pigmentation and growth rates of isolates, single-conidium subcultures were grown on fresh PDA dishes and incubated under alternating dark and light with a 12-h photoperiod at 25 °C for two weeks. For microscopic examination, all strains were grown on carnation leaf-piece agar (CLA) [48,49], potassium chloride agar (KCL-Agar) [50], and spezieller nahrstoffarmer agar (SNA) [51] Petri dishes, which were incubated at 25 °C for 4 days (KCL-Agar dishes) and also for 14 days (CLA and SNA dishes) under alternating dark and light with a 12-h photoperiod. Colony and conidia characteristics including the colony growth rate, pigmentation, mean size of 30 randomly selected well-developed macroconidia, and number of septa were recorded. Finally, the descriptions by Summerbell and Schroers [33], Leslie and Summerell [52], Nalim et al. [28], and Short et al. [32] were used for morphological species determination.

### 2.3. Pathogenicity Tests

Overall, 25 isolates were evaluated for their pathogenicity on their host of origin. Moreover, four of these isolates, Iv-Km50, Iv2r30, Far-317, and FS-Spa, were assessed for their pathogenicity on 11 different cucurbit varieties and species, including different varieties of melon, i.e., Garmak-Ahmadabadi (*Cucumis melo* var. *reticulatus*), Shahd-e-Shiraz (*C*. *melo* var. *cantalopensis*), Kharboz-e-Mashhadi (*C*. *melo* var. *indorous*), Snake melon (*C*. *melo* var. *flexusus*), Til-Mashhad (*C*. *melo* L.), and Semsoori (*C*. *melo* L.), as well as five other species of cucurbits, cucumber (*C*. *sativus* L.), watermelon (*Citrullus lanatus* L.), zucchini (*Cucurbita pepo* L.), pumpkin (*C*. *moschata* L.), and bottle gourd (*Lagenaria siceraria* L.). Inoculum was prepared by adding five discs (4 mm diameter) from 7-day-old *Fusarium* colonies grown on PDA to flasks containing sterile wheat seeds as described by Sabahi et al. [13]. Sterile agar plugs were added to wheat seeds to inoculate control plants. Flasks were incubated at room temperature for 15 days to ensure complete colonization of the grains. Inoculation of 15-day-old cucurbit seedlings was performed using a procedure described previously [13,53]. Inoculated plants were kept under greenhouse conditions (25 °C and 65% relative humidity) for symptoms’ development until 30 days post-inoculation (dpi). To determine the virulence and pathogenicity of FSSC isolates, the seedlings were examined 30 dpi; they were cut at cotyledon level, and the symptom severity (SS) was scored according to a scale from 0 to 5 as described by Nagao et al. [54]. Nine plants of each host were inoculated per fungal isolate, and the same number of plants inoculated with water was used as a control. SS measurements were converted from original ordinal scale to ratio scale and normalized (from 0 to 1) using the following formula:SS = [(0 × n_0_) + (1 × n_1_) + (2 × n_2_) + (3 × n_3_) + (4 × n_4_) + (5 × n_5_)]/N × M_i_
where n_0_, n_1_, n_2_, n_3_, n_4_, and n_5_ are the number of symptomatic plants per each scale level (from 0 to 5), N is total number of plants examined, and M_i_ is the highest score scale. SS data were analyzed using analysis of variance (ANOVA), and differences between the strains were analyzed using the GLM procedure of the SPSS software (SPSS, version 16). Tukey’s test was used for pairwise mean comparisons.

After symptom scoring, re-isolation was performed from symptomatic seedlings using PDA. The fungal isolates were identified based on morphological characteristics and sequencing of three genetic regions (ITS, LSU, and *tef1*) to fulfill Koch’s postulates. The experiments were repeated once with similar results.

### 2.4. DNA Extraction, PCR Amplification and Phylogenetic Analysis

Genomic DNA extraction and purification were performed using the procedures described by Sabahi et al. [10]. Internal transcribed spacer (ITS), nuclear large-subunit (28S) rDNA (LSU), and translation elongation factor 1-alpha (*tef1*) gene were amplified using the primer pairs ITS1/ITS4 [55], NL1/NL4, [56], and EF1-728F/EF2 [57], respectively. For PCR reactions, the Universal PCR Kit Ampliqon^®^ Taq DNA Polymerase Master Mix Red (Ampliqon A/S, Odense, Denmark) was used according to the manufacturer’s recommendations. For each fungal isolate, a 50 µL PCR reaction including 50 ng total DNA and 1 µL of each primer (10 pmol µL^−1^) was used. The PCR amplification conditions were 30 cycles of 95 °C for 5 min, 95 °C for 50 s, 54 °C for 1 min, 72 °C for 50 s, and final step of 72 °C for 10 min. PCR products were sent to Bioneer Corporation (http://www.Bioneer.com (accessed on 4 March 2023)) to be sequenced via Sanger sequencing technology.

Newly obtained sequences were blasted against databases available at BLAST [58] on NCBI-GenBank database. Three loci in a set of worldwide FSSC strains were retrieved from the GenBank database and included in the phylogenetic analysis (Appendix A). Sequences were aligned using the CLUSTAL W program, and concatenated following the alphabetic order of the genes, ending in a sequence of 1778 base pairs: nucleotides 1 to 667 for ITS, 668 to 1111 for LSU, and 1112 to 1778 for *tef1*. 

Phylogenetic trees were constructed using the concatenated sequences of three loci via maximum likelihood with MEGA 6.06 [59]. The best model of evolution was determined using the Modeltest option from MEGA 6.06, and the phylogenetic tree was constructed with bootstrapping (1000 replications)*. Fusarium staphyleae* (NRRL 22316) was used as an outgroup, and the final tree was drawn using infix pdf editor [60].

### 2.5. Genetic Diversity and Haplotype Network Analysis

Nucleotide diversity, number of haplotypes, haplotype frequency, haplotype diversity, number of segregating sites, number of mutations, percentage of polymorphism site, and the minimum number of recombination events were estimated using DnaSP v. 5.10 software for the sequences of each gene as well as concatenated sequences of FSSC isolates. The class I neutrality test (Tajima’s D, Fu and Li’D^*^, and Fu and Li’s F^*^ statics) were also calculated for detecting departure from the mutation-drift equilibrium [61]. NeighborNet networks were constructed, and the pairwise homoplasy index (PHI-Test) was estimated for each gene region and the combined data set by SplitsTree v. 4.18.2 [62]. To see the phylogeographic relationship between the FSSC isolates, a haplotype network was generated for the concatenated sequences data set using TCS (Tata Consultancy Services) algorithm [63], which is implemented in Population Analysis with Reticulate Trees (PopART v. 1.7) [64]. The geographic origin of FSSC isolates investigated in this study and the FSSC strains retrieved from the GenBank database (Table 1) were allocated into the haplotype network as described by Leigh and Bryant [64].

## 3. Results

### 3.1. FSSC Isolates Obtained from Symptomatic Melon Plants in Iran

Melon growing regions in five provinces of Iran, including Fars, Isfahan, Khorasan, Semnan, and Yazd, were surveyed for the occurrence of Fusarium wilt. Melons with the Fusarium wilt syndrome consisting in wilt associated to crown and root rot were observed in all surveyed provinces (Figure 1).

A total of 41 *Fusarium*-like isolates were recovered from symptomatic melon plants of two varieties, *C. melo* L. var. *cantalopensis* Naudin (cantaloupe) and *C. melo* L. var. *inodorous* Naudin (long melon). All isolates were morphologically identified as *F. solani sensu* Leslie and Summerell [51]. Among the FSSC isolates examined in this study, 15 and 10 were from Khorasan and Semnan provinces, respectively, seven and five from Fars and Yazd provinces, respectively, and four from Isfahan province.

Based on the geographical origins, host, and morphological characteristics, 25 isolates representing the overall diversity of the original set of isolates were selected for in depth investigation (Table 1). An isolate of FSSC from Spain (FS-Spa) was included in this study as a reference isolate. According to the sequencing data of three genetic regions (ITS, LSU, and *tef1*) and morphological characteristics, four phylogenetic species were identified among the 25 Iranian isolates (Table 1 and Table 2, Appendix A). Of these, *F. falciforme* (syn. *Neocosmospora falciformis*) (18 isolates), *F. keratoplasticum* (syn. *N. keratoplastica*) (a single isolate), and *F. vanettenii* (syn. *N. pisi*) (five isolates) are known species. One isolate and the reference isolate from Spain were of an undescribed phylogenetic species of *Fusarium sensu lato*, tentatively named FSSC 5 or *Neocosmospora* sp.

Significant differences between distinct phylogenetic species were noticed in some morphometric and cultural characteristics, such as the shape of macroconidia and the growth rate on PDA (Table 2). In particular, the shape index (length to width ratio) of macroconidia of *F. falciforme* isolates ranged from 7.9 to 8.3, with a mean ± SD of 8.1 ± 0.12, while the same index for macroconidia of *F. venattenii* isolates ranged from 10.2 to 10.6, with a mean ± SD of 10.4 ± 0.23. The value of shape index of macroconidia of the only *F. keratoplasticum* isolate was 7.9, while the corresponding values for the two FSSC 5 isolates (Tay-r2-r, from Iran, and FS-Spa, from Spain) were 7.9 and 7.0, respectively. The growth rate on PDA of *F. falciforme* isolates ranged from 7.0 to 9.0 mm/day, with a mean ± SD of 8.1 ± 0.75, while the growth rate of *F. venattenii* isolates ranged from 5.0 to 6.0 mm/day, with a mean ± SD of 5.6 ± 0.42 mm/day. The growth rate of the *F. keratoplasticum* isolate was 8.5 mm/day, while the corresponding values for the two FSSC 5 isolates (Tay-r2-r, from Iran, and FS-Spa, from Spain) were 8.5 and 8.0 mm/day, respectively. Other morphological characteristics overlapped among different phylogenetic species or were not enough discriminant, as they showed a great intraspecific variability.

### 3.2. Pathogenicity and Host Range of FSSC Strains

Under greenhouse conditions, all 25 Iranian isolates and the reference isolate from Spain evaluated in this study were shown to be pathogenic on the host plant from which they were isolated; symptoms of wilting and crown- and root-rot were observed in artificially inoculated plants (Appendix A). In the host range assays, performed with four strains, each representing a distinct phylospecies, all melon varieties were severely affected by Iv2-r-30 and Iv-km-50 isolates, with SS values around 1, while the other cucurbit crops were less severely affected with the least severe symptoms being observed on zucchini plants with an SS value of 0.7 (Figure 2a). 

The symptoms induced by Far-317 and FS-Spa isolates on six varieties of melons were less severe than those induced by Iv2-r-30 and Iv-km-50 isolates. Conversely, the degree of aggressiveness of FS-Spa strain on cucumber, watermelon, zucchini, pumpkin, and bottle gourd plants was higher than the other FSSC isolates evaluated in this study. Among all four isolates tested, Far-317 was the least aggressive on cucurbits (Figure 2b).

The inoculated fungi were re-isolated from symptomatic, artificially infected plants on PDA medium and their identity was confirmed by their morphological characteristics on SNA, CLA, and KCL-Agar media, as well as the sequencing of three genetic regions (ITS, LSU, and *tef1*). The negative (mock-inoculated) control plants remained healthy and did not develop any symptoms. The same results were observed in a replication of the pathogenicity and host range assays.

### 3.3. Phylogenetic Analyses

Sequence analysis of three genetic regions (ITS, LSU, and *tef1*) was conducted on all 25 selected Iranian FSSC isolates and the reference isolate from Spain. The nucleotide sequences of FSSC isolates were deposited in GenBank database (see Table 1).

Based on BLASTn searches using the sequence of *tef1* gene on the NCBI GenBank database, the 26 strains were identified as either *F. falciforme*, *F. keratoplasticum*, *F. vanettenii*, or an undescribed species belonging to the FSSC. Overall, 17 of the 25 isolates from Iran clustered with the *F. falciforme* group with high bootstrap support (86%). *Fusarium falciforme* strains are divided into three subclusters, two of which, subcluster I and III, were found in Iran (Figure 3). The isolates of subcluster I showed two and three nucleotide differences in the sequences of ITS and *tef1* gene regions, respectively, compared to the sequence of the subcluster III isolates. However, no nucleotide differences in the sequences of the LSU region between isolates of subclasters I and III were observed. The phylogenetic tree further showed a strongly-supported relationship (96% bootstrap support) between *F. vanettenii* (NRRL 22820 and NRRL 22278) obtained from GenBank and six isolates from melon plants in Iran. In addition, the tree showed that one isolate from Iran (Tay-r2-r) and the reference isolate from Spain (FS-Spa) belong to the undescribed species FSSC 5 (*Neocosmospora* sp.) with high bootstrap value (95%), while another Iranian isolate (Iv-km-50) was placed in *F*. *keratoplasticum* with strong phylogenetic affinity (100% bootstrap support) (Figure 3). Similar results were obtained when the sequences of gene *tef1* were analyzed separately (Appendix A), while results of the phylogenetic analysis of the ITS and LSU regions were not consistent (Appendix A). Based on ITS maximum likelihood phylogeny, all *F. falciforme* isolates were placed in one group, while *F. solani* f. sp. *robiniae* and *F. petroliphilum* isolates from GenBank clustered together. As well, *F. vanettenii* and *F. solani* f. sp. *mori* isolates retrieved from GenBank clustered in one group, and Iranian isolates clustered in a separate group (Appendix A). In the LSU-based maximum likelihood phylogeny, all *F. falciforme* isolates clustered together, while *F. vanettenii* isolates clustered with the *F. solani* f. sp. *mori* and *F. solani* f. sp. *robiniae* isolates retrieved from GenBank (Appendix A).

### 3.4. Genetic Diversity

The FSSC isolates recovered from melons in Iran carried different allelic forms and sequence types (STs) and corresponded to 10 multilocus haplotypes (MHs) based on the concatenated sequences of the three gene regions. Six, one, two, and one MHs belonged to *F. falciforme*, *F. keratoplasticum*, *F. vanettenii*, and FSSC 5 isolates, respectively (Table 1; Figure 4). As for the individual gene regions, 4, 10, and 5 STs were detected for the ITS, *tef1* and LSU, respectively, in the Iranian isolates while 8, 13, and 4 STs were identified for these gene regions in the FSSC isolates from other countries (Table 1; Appendix A). Sequence variation statistics and diversity parameters were estimated by DnaSP v. 5.10 software for three individual gene regions, as well as concatenated sequences in all evaluated FSSC isolates in this study (Table 3). Similarly, the diversity parameters were calculated among the *F. falciforme* isolates recovered in Iran and compared with *F. falciforme* isolates from other countries (Table 3). As for the *F. falciforme* isolates from Iran, there were 1, 6, and 2 STs in the sequences of ITS, *tef1*, and LSU gene regions, respectively, while 1, 1, and 1 STs were found among the *F. keratoplasticum* isolates, 1, 2, and 1 STs among the *F. vanettenii* isolates, and 1, 1, and 1 STs among the isolates of the undescribed species FSSC 5 for the sequences of ITS, *Tef-1α*, and LSU gene regions, respectively (Table 1; Appendix A). Based on the concatenated sequences of three gene regions, the haplotype frequency (HF) and haplotype diversity (HD) indices were 0.352, and 0.721, respectively, for *F. falciforme* isolates from Iran and 0.857, and 0.952, respectively, for *F. falciforme* isolates from other countries, indicating higher genetic diversity among isolates from other countries (Table 3).

Most isolates from different regions of Iran were placed in separate MHs compared to isolates from other countries. Only one isolate, Iv-km-50 (*F. keratoplasticum*), recovered in Khorasan (Iran), shared the same MH as isolate NRRL 32780 from USA (Figure 4). Some MHs were represented by several Iranian isolates. For instance, isolates Se-r-19, Iv-k-21, Yazd-m-23, Tj-90, Tj-3, Kht-r-f1, Kno-2, and Kho-r2-b of *F. falciforme* belonged to the same MH. Conversely, several MHs were unique and represented by only a single Iranian isolate. This was the case of isolate Tk-rs-1, Khaf-400, Ka-s-82, and Ga-r-30 of *F. falciforme*, and isolate Tay-r2-r of FSSC 5 (Figure 4; Table 1).

A phylogenetic network was constructed by the NeighborNet method using the concatenated sequences of three gene regions. While the minor reticulations observed in the NeighborNet network indicate possible recombination events within the Iranian FSSC population, the PHI-Test did not find statistically significant evidence for recombination either for each gene region or in concatenated gene regions. The results of PHI-Test were in general accordance with those of the DnaSP results and did not show statistically significant evidence for recombination. Population neutrality indices (i.e., Fu and Li’ *D**, and Fu and Li’s *F**) were significantly negative for the LSU gene region of *F. falciforme* isolates (Table 3), indicating a recent selective sweep and/or population expansion after a recent bottleneck. However, considering the nonsignificant results obtained in sequences of ITS, and *tef1* gene regions, further investigation by sequences of additional gene regions is needed to confirm these observations.

## 4. Discussion

This study discloses the genetic variability, potential host range, and geographic distribution of the FSSC population associated with Fusarium wilt of melon in Iran. Field surveys for three consecutive years (2009–2011) showed the disease occurred across five major melon-producing provinces of the country. Both molecular and morphological data were used to identify species of the FSSC recovered from symptomatic melon plants. In an earlier phylogenetic study, it was shown that DNA sequences of the LSU, ITS and *tef1* gene regions can resolve evolutionary relationships within the FSSC [28]. Therefore, in the present study the combined data set of these three loci was used to identify FSSC isolates from melon at the species level. FSSC strains have been previously divided into three distinct subgroups, termed clades 1, 2, and 3 [25,26,27]. All Iranian FSSC isolates from melons, as well as the FS-Spa reference isolate from Spain, were found to be members of clade 3. Phylogenetic analyses of the concatenated three gene regions revealed that Iranian isolates grouped into five lineages, three of which had been identified earlier as *F. vanettenii*, *F. keratoplasticum,* and the undescribed species FSSC 5. According to the taxonomic criterion proposed by Sandoval-Denis and Crous [36] and Sandoval et al. [37], assigning the whole FSSC to *Neocosmospora* as a genus distinct from *Fusarium* s.s., the three above mentioned species should be named *N. pisi*, *N. keratoplastica,* and *Neocosmospora* sp., respectively, while *F. falciforme* is a synonym of *N. falciformis*. Here, both nomenclatural criteria have been interchangeably adopted to bypass the dispute concerning the taxonomy of the genus *Fusarium* and to make it easier to compare results from studies of diverse authors. The majority of Iranian FSSC isolates from melon fell into *F. falciforme* (FSSC 3 + 4) (syn. *N. falciformis*) and grouped into two subclusters. Similar results were obtained when sequences of *tef1* were used individually for phylogenetic analyses, while all Iranian *F. falciforme* isolates were grouped into only one cluster based on phylogenetic analysis of ITS and LSU sequences. In ITS and LSU-based phylogenetic analysis, the *F. vanettenii* isolates clustered together with the isolates of *F. solani* f. sp. *mori* and *F. solani* f. sp. *robiniae*, indicating these gene regions were not suitable to distinguish these lineages.

The *F. falciforme* and *F. vanettenii* phylogenetic species showed distinct cultural and morphometric characteristics, such as the growth rate on agar medium and the shape of macroconidia. Conversely, no significant difference in these morphological traits was noticed between the *F. falcforme* isolates and either the *F. vanettenii* or the FSSC 5 isolates. 

All FSSC isolates evaluated in this study were pathogenic on melon plants. Although *N. keratoplastica* and *N. falciformis* have been reported previously to cause root rot of muskmelons in Spain [65,66], to our knowledge, this is the first report of *F. vanettenii* and FSSC 5 causing wilt and root rot of melons in Iran. *Fusarium vanettenii* (synonym: *Neocosmospora pisi*), formerly *Fusarium solani* f. sp. *pisi*, is a recently recognized species in the FSSC [38]. This fungus has been reported as a destructive pathogen of legumes in different parts of the world, including Canada, Czech Republic, India, Iran, New Zealand, Southern Scandinavia, United Kingdom, and USA [47,67]. Šišić et al. [68] demonstrated this pathogen has a broad host range and raised doubts about the definition of it as a *forma specialis*. Consistently with this hypothesis, *F. vanettenii* was recently reported as a causal agent of tomato root rot in India [69]. Results of the present study confirm the host range of this species also encompasses non-leguminous hosts. It is the first time *N. pisi* is reported as a pathogen of melon worldwide. Members of FSSC 5 have been mostly reported as clinical opportunistic pathogens associated with infectious diseases of humans and animals [22]. Moreover, this phylogenetic species is a soil inhabitant and has been reported to be associated to the dry rot of potato [31,70]. In the present study, the FSSC 5 isolates were proved to be pathogenic on melon plants and other cucurbits, further expanding the known, already broad, host range of this *Fusarium* lineage. The melon varieties tested in this study, all widely grown in Iran, were susceptible to FSSC isolates. In greenhouse assays, besides melon, other crops of the *Cucurbitaceae* family, including watermelon, cucumber, zucchini, pumpkin, and bottle gourd, showed wilting and root rot symptoms when inoculated with FSSC isolates recovered from melon plants with natural infections. Although there was a noticeable variability in virulence among the isolates, no differences were observed in their host range on cucurbits. 

FSSC species are considered cosmopolitan pathogens, as they occur in all climatic regions [71,72]. However, they prefer tropical hot areas [73,74]. In all provinces surveyed in this study, melon crops were in plains with warm to hot springs and summers and were planted in spring (March to April), so climatic conditions were relatively uniform. Despite this, the incidence of the disease was higher in Khorasan and Semnan. Moreover, a certain geographical structure in the Iranian FSSC population associated to Fusarium wilt of melon seems to exist. However, these aspects need to be further investigated to be confirmed.

A total of 24 MHs were identified among the FSSC isolates analyzed in this study; nine of them were only found in Iranian isolates, and one was shared between Iranian isolate and isolates from other country. Haplotypes MH2 and MH22 were only found in the Khorasan province, the first producer of melon in Iran with more than 48% of the national production [75].

Overall, 12 MHs were found among the isolates of *F. falciforme* (syn. *N. falciformis*). According to Posada and Crandall [76], the most frequent haplotype is probably the oldest in a given population. MH4 was the most common and widely distributed haplotype among Iranian *F. falciforme* isolates and included eight strains from five different provinces. The wide geographic distribution of this haplotype suggests effective mechanisms of dispersion over long distances. Very probably, in melon crops fungi of the FSSC have been transmitted prevalently by seeds, as *F. solani s.l.* is known to be a seed-borne pathogen [77]. A patchy distribution of symptomatic plants in surveyed melon crops, which is typical of seed-borne pathogens, would reinforce this hypothesis. 

Interestingly, haplotype MH13 included two isolates of *F. keratoplasticum* (syn. *N. keratoplastica*) recovered from melon in Iran and sea turtles in the USA. This is not so surprising as several taxa of the FSSC have been associated with both clinical infections and plant diseases [41,65]. For instance, *N. keratoplastica*, a relevant pathogen of animals (including humans), has been recently reported as causal agent of wilt and root rot of muskmelon and watermelon crops [65]. Similarly, *N. petroliphila* (syn. *F. petroliphilum*), responsible for human keratitis, has been formerly known among plant pathologists as *F. solani* f. sp. *cucurbitae* race 1, a causal agent of fruit, stem, and root rot of cucurbits [65,78]. *Neocosmospora falciformis* (syn. *F. falciforme*), which is frequently responsible for clinical infections on humans, has been reported as a pathogen of several host plants, including species of the *Cucurbitaceae* family, confirming a wide host and ecological range of members of the FSSC [41,66].

## 5. Conclusions

This study provides preliminary information on the genetic variability of FSSC populations associated with Fusarium wilt of melon in Iran. Four phylogenetic species, including *N. falciformis* (syn. *F. falciforme*), *N. pisi* (syn. *F. vanettenii*), *N. keratoplastica* (syn. *F. keratoplasticum*), and *Neocosmospora* sp. (FSSC 5), and 10 diverse haplotypes have been identified in a set of isolates collected in major melon producing provinces of the country. Isolates of diverse genotypes differed in virulence, but all were pathogenic on the most common melon varieties grown in Iran and on a wide range of other cucurbits. The diversity of FSSC genotypes associated with Fusarium wilt of melon as well as their broad host range and ecological plasticity have implications for epidemiology, disease management strategies, breeding programs for disease resistance, and quarantine measures. The study of the genetic variability of FSSC populations would benefit from the use of additional markers. Moreover, targeting pathogenesis-related genes could provide a better insight into the biology and epidemiology of members of this complex. Focusing on taxa with a very broad host range that comprises both plant and animal pathogens might be helpful to understand the pathogenesis mechanisms of these fungi and in particular genetic determinants of both their polyphagia and ability to switch from a saprophytic to a parasitic lifestyle.

## Figures and Tables

**Figure 1 jof-09-00486-f001:**
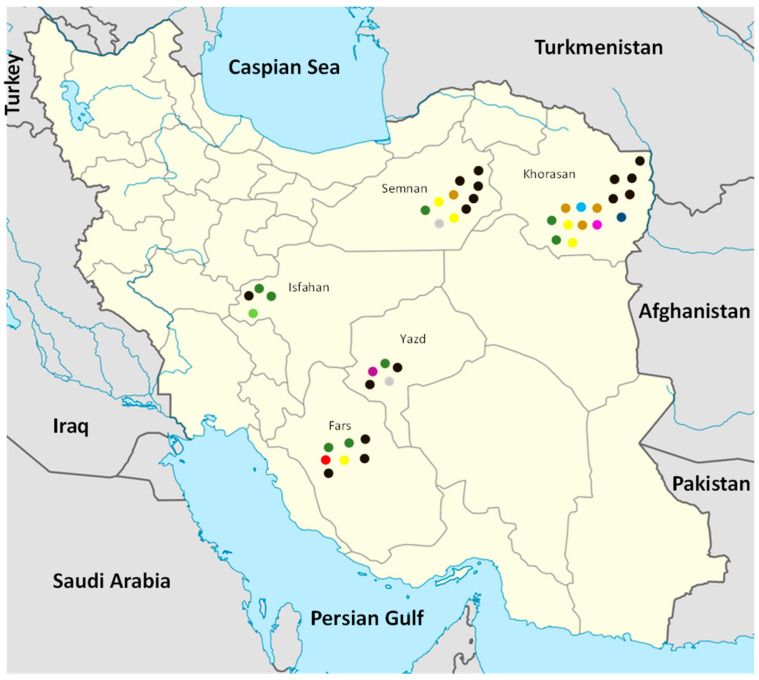
Distribution of FSSC isolates recovered from melons in Iran and examined in this study. Each isolate was recovered from a distinct field. In the map, each dot represents an isolate and each color represents a multilocus haplotype (MH). Dark green: MH 4; Light green: MH 12; Dark blue: MH 2; Yellow: MH 10; Pink: MH 22; Brown: MH 16; Light blue: MH 13; Red: MH 11; Gray: MH 15; Purple: MH 3.

**Figure 2 jof-09-00486-f002:**
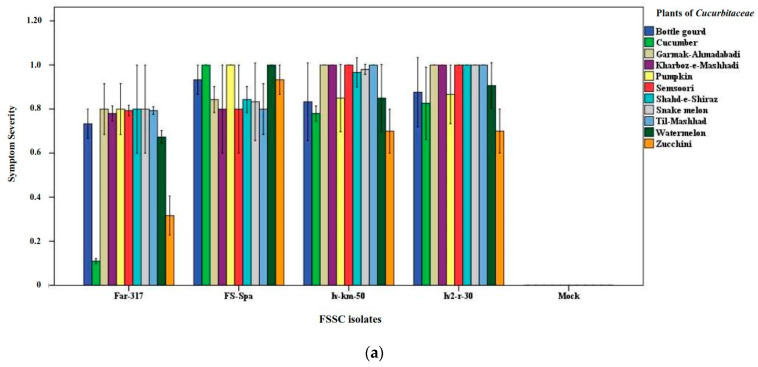
Results of pathogenicity assays of four isolates of diverse FSSC species on different cucurbits. The FSSC species tested included *F. keratoplasticum* (isolate Iv-km-50), *F. falciforme* (isolate Iv2-r-30), *F. vanettenii* (isolate Far-317), and FSSC 5 (isolate FS-Spa). Nine plants of each cucurbit species or variety were inoculated per fungal isolate, and the same number of plants was mock-inoculated with water. Symptom severity was evaluated 30 days after inoculation. (**a**) Symptom severity induced by each *Fusarium* isolate on 11 diverse cucurbits (means of nine values ± SE). (**b**) Values of symptoms severity induced by each *Fusarium* isolate on 11 diverse cucurbits were pooled together to compare the virulence of isolates (means of 99 values ± SE).

**Figure 3 jof-09-00486-f003:**
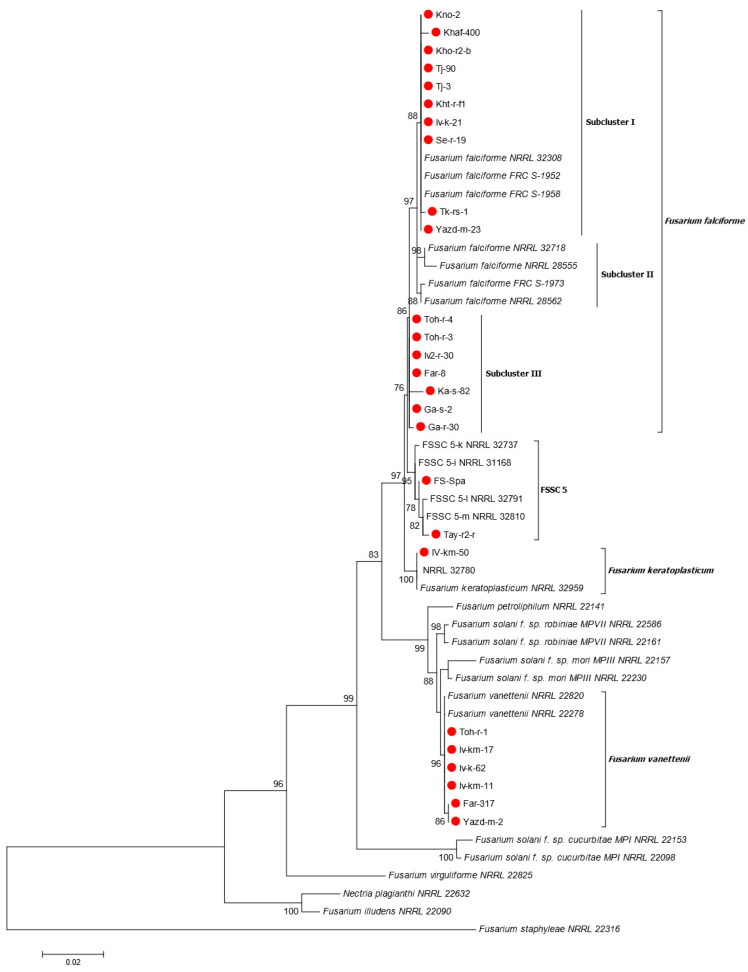
Maximum likelihood phylogeny of FSSC isolates from melons based on concatenated sequencing data of three gene regions (ITS, LSU, and *tef1*). Scale bar indicates number of substitutions per site. Bootstrap values higher than 70 are shown. *Fusrium staphyleae* (NRRL 22316) was used as the outgroup. Red dots indicate Iranian isolates recovered in this study.

**Figure 4 jof-09-00486-f004:**
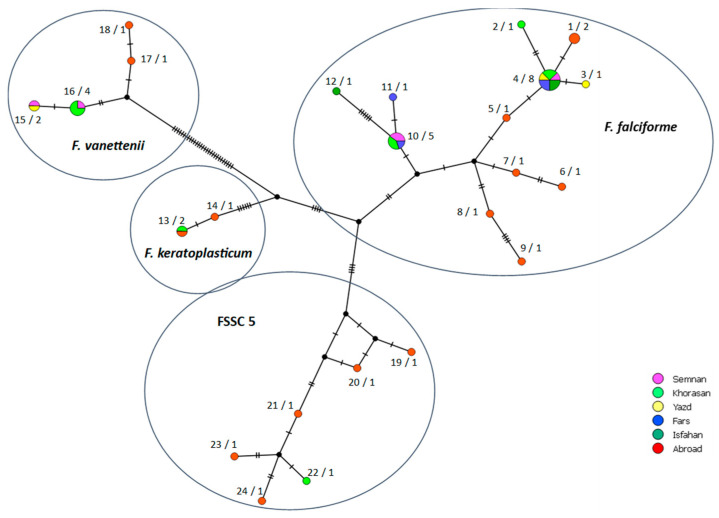
TCS multilocus haplotype network generated using the POPArt program by combining three gene regions (ITS, LSU, and *tef1*) of FSSC isolates recovered from melons in Iran. The size of the circles indicates the relative frequency of sequences of a given multilocus haplotype (MH). Hatch marks along the branches indicate the number of mutations. Each color represents one of the five provinces where the FSSC isolates were recovered. The isolates retrieved from GenBank and FS-Spa reference isolate received from Spain are assigned to the category ‘Abroad’. The numbers on the left of a slash indicates the MH number (Table 1), while the numbers on the right of the slash indicates the number of isolates in a given MH.

**Table 1 jof-09-00486-t001:** FSSC isolates recovered from melons in Iran, their host and geographic area of origin, date of isolation, and multilocus haplotypes (MHs) and sequence types (STs) based on the partial sequences of three gene regions (ITS, LSU, and *tef1*).

Isolate	Species	Year ^a^	Source	Collection ^b^ Locality	Sequence Type (ST) ^c^ Detected in:			Multilocus Haplotype (MH)
					ITS	LSU	*tef1*	
Se-r-19	*F. falciforme*	2010	long melon	Se, Is, Ir	7	2	12	4
Khaf-400	*F. falciforme*	2009	long melon	Kha, Kho, Ir	7	2	13	2
Toh-r-3	*F. falciforme*	2009	long melon	Toh, Kho, Ir	7	2	6	10
Toh-r-4	*F. falciforme*	2009	long melon	Toh, Kho, Ir	7	2	6	10
Iv-k-21	*F. falciforme*	2011	long melon	Ey, S, Ir	7	2	12	4
Iv2-r-30	*F. falciforme*	2009	long melon	Ey, S, Ir	7	2	6	10
Yazd-m-23	*F. falciforme*	2010	cantaloupe	Mey, Y, Ir	7	2	12	4
Tj-90	*F. falciforme*	2009	long melon	Toj, Kho, Ir	7	2	12	4
Tk-rs-1	*F. falciforme*	2011	cantaloupe	Ta, Y, Ir	7	2	14	3
Ga-r-30	*F. falciforme*	2009	cantaloupe	Ga, Fa, Ir	7	2	7	11
Ga-s-2	*F. falciforme*	2011	cantaloupe	Ga, Fa, Ir	7	2	6	10
Ka-s-82	*F. falciforme*	2010	long melon	Ka, Is, Ir	7	1	5	12
Tj-3	*F. falciforme*	2010	long melon	Toj, Kho, Ir	7	2	12	4
Far-8	*F. falciforme*	2011	long melon	Fa, S, Ir	7	2	6	10
Kht-r-f1	*F. falciforme*	2009	long melon	Khan, Is, Ir	7	2	12	4
Kno-2	*F. falciforme*	2009	cantaloupe	Khaf, Fa, Ir	7	2	12	4
Kho-r2-b	*F. falciforme*	2011	cantaloupe	Khaf, Fa, Ir	7	2	12	4
Tay-r2-r	FSSC 5	2009	long melon	Tay, Kho, Ir	1	3	18	22
Iv-k-62	*F. vanettenii*	2009	long melon	Ey, S, Ir	3	5	2	16
Far-317	*F. vanettenii*	2009	long melon	Far, S, Ir	3	5	1	15
Iv-km-11	*F. vanettenii*	2010	long melon	Kash, Kho, Ir	3	5	2	16
Iv-km-17	*F. vanettenii*	2010	long melon	Kash, Kho, Ir	3	5	2	16
Toh-r-1	*F. vanettenii*	2009	long melon	Toh, Kho, Ir	3	5	2	16
Yazd-m-2	*F. vanettenii*	2010	long melon	Mey, Y, Ir	3	5	1	15
Iv-km-50	*F. keratoplasticum*	2009	long melon	Kash, Kho, Ir	6	4	3	13
Strains from different countries and sources								
NRRL 32718	*F. falciforme*	2006	human eye	USA	7	2	10	8
FRC S-1973	*F. falciforme*	2011	soil	Aust	7	2	9	6
FRC S-1958	*F. falciforme*	2011	soil	Aust	9	2	12	1
FRC S-1952	*F. falciforme*	2011	soil	Aust	9	2	12	1
NRRL 28555	*F. falciforme*	2006	human wrist	USA	8	2	11	9
NRRL 28562	*F. falciforme*	2006	human bone	USA	7	2	8	7
NRRL 32308	*F. falciforme*	2006	human foot	SA	7	2	12	5
FS-Spa	FSSC 5	ND	ND	Spain	1	3	19	23
NRRL 31168	FSSC 5	2006	human toe leukemia	USA	2	3	15	20
NRRL 32810	FSSC 5	2006	human eye	USA	1	3	16	21
NRRL 32737	FSSC 5	2006	human eye	USA	2	3	20	19
NRRL 32791	FSSC 5	2006	human	USA	2	3	17	24
NRRL 22820	*F. vanettenii*	ND	ND	USA	5	5	2	18
NRRL 22278	*F. vanettenii*	ND	ND	USA	4	5	2	17
NRRL 32780	*F. keratoplasticum*	2006	sea turtle	USA	6	4	3	13
NRRL 32959	*F. keratoplasticum*	2006	human skin	USA	6	4	4	14

^a^ ND: not determined. ^b^ Se: Sefidshahr; Is: Isfahan; Ir: Iran; Kha: Khaf; Kho: Khorasan; Toh: Torbat-e Heydariyeh; Ey: Eyvanekey; S: Semnan; Mey: Meybod; Y: Yazd; Toj: Torbat-e Jam; Ta: Tabas; Ga: Galehdar; Fa: Fars; Ka: Kashan; Far: Faravan; Khan: Khansar; Khaf: Khafr; Tay: Taybad; Kash: Kashmar; Aust: Australia; SA: Saudi Arabia. ^c^ Sequence type is based on TCS sequence type network as shown in Appendix A.

**Table 2 jof-09-00486-t002:** Morphological characteristics of FSSC isolates obtained from melon plants in Iran.

Isolate	Species	Colony Growth Rate (mm/day)	Pigment on PDA ^a^	Chlamydospores ^b^	Shape/No. of Septa of Microconidia ^a^	Shape of Basal and Apical Cell ^a^	Length × Width of Macroconidia (µm) ^c^	
							3 and 4 Septate	5 Septate
Se-r19	*F. falciforme*	7.5	WY or G	+	EO, OT/0-1	BNP, Cu	40.5 ± 1.5 × 5.0 ± 0.5	-
Khaf-400	*F. falciforme*	7.5	WY or G	+	EO, OT/0-1	BNP, Cu	41.5 ± 2.5 × 5.1 ± 0.5	-
Toh-r3	*F. falciforme*	9.0	WY or G	+	EO, OT/0-1	BNP, Cu	42.5 ± 2.5 × 5.2 ± 0.5	-
Yazd-m23	*F. falciforme*	8.0	WY or G	+	EO, OT/0-1	BNP, Cu	41.0 ± 1.5 × 5.2 ± 0.5	-
Tj-90	*F. falciforme*	8.5	WY or G	+	EO, OT/0-1	BNP, Cu	42.0 ± 2.5 × 5.1 ± 0.5	-
Tk-rs-1	*F. falciforme*	9.0	WY or G	+	EO, OT/0-1	BNP, Cu	41.5 ± 2.5 × 5.0 ± 0.5	-
Ga-r30	*F. falciforme*	9.0	WY or G	+	EO, OT/0-1	BNP, Cu	40.5 ± 1.5 × 5.0 ± 0.5	-
Ga-s2	*F. falciforme*	8.5	WY or G	+	EO, OT/0-1	BNP, Cu	42.5 ± 2.5 × 5.2 ± 0.5	-
Ka-s-82	*F. falciforme*	9.0	WY or G	+	EO, OT/0-1	BNP, Cu	40.5 ± 1.5 × 5.0 ± 0.5	-
Tj-3	*F. falciforme*	7.0	WY or G	+	EO, OT/0-1	BNP, Cu	41.5 ± 2.5 × 5.1 ± 0.5	-
Far-8	*F. falciforme*	7.5	WY or G	+	EO, OT/0-1	BNP, Cu	41.5 ± 2.5× 5.1 ± 0.5	-
Yazd-m2	*F. falciforme*	8.0	WY or G	+	EO, OT/0-1	BNP, Cu	42.0 ± 2.5 × 5.3 ± 0.5	-
Kho-r2-b	*F. falciforme*	8.0	WY or G	+	EO, OT/0-1	BNP, Cu	42.5 ± 2.5 × 5.1 ± 0.5	-
Kht-r-f1	*F. falciforme*	7.5	WY or G	+	EO, OT/0-1	BNP, Cu	42.5 ± 2.5 × 5.2 ± 0.5	-
Toh-r4	*F. falciforme*	7.0	WY or G	+	EO, OT/0-1	BNP, Cu	40.5 ± 1.5 × 5.0 ± 0.5	-
iv-k-21	*F. falciforme*	9.0	WY or G	+	EO, OT/0-1	BNP, Cu	40.5 ± 1.5 × 5.0 ± 0.5	-
iv2-r30	*F. falciforme*	8.5	WY or G	+	EO, OT/0-1	BNP, Cu	41.5 ± 2.5 × 5.1 ± 0.5	-
Kno2	*F. falciforme*	7.0	WY or G	+	EO, OT/0-1	BNP, Cu	41.0 ± 2.5 × 5.2 ± 0.5	-
iv-km-50	*F. keratoplasticum*	8.5	WY	+	O, PC, S/0-1	NB	36.5 ± 2.3 × 4.8 ± 0.3	-
Far-317	*F. venattenii*	5.5	WB	+	OR and T/0-1	PR	46.2 ± 2.0 × 4.35 ± 0.4	-
iv-km-11	*F. venattenii*	6.0	WB	+	OR and T/0-1	PR	45.2 ± 1.5 × 4.35 ± 0.4	-
iv-km-17	*F. venattenii*	6.0	WB	+	OR and T/0-1	PR	46.2 ± 2.5 × 4.33 ± 0.1	-
Toh-r1	*F. venattenii*	5.0	WB	+	OR and T/0-1	PR	44.1 ± 2.3 × 4.34 ± 0.2	-
iv-k-62	*F. venattenii*	5.5	WB	+	OR and T/0-1	PR	44.2 ± 2.0 × 4.35 ± 0.3	-
Tay-r2-r	FSSC 5	8.0	WY or G	+	EO, OT/0-1	BNP, Cu	40.5 ± 1.5 × 5.1 ± 0.5	-
FS-Spa	FSSC 5	8.5	WC	+	O, EO, C, R/0-1	BN, P Cu	38.6 ± 1.5 × 5.5 ± 0.3	42.5 ± 2.5 × 5.8 ± 0.3

^a^ WY: white to yellow; G: green; WB: white to brown; WC: white to cream; EO: elongated oval; OT: obovoid with a basal truncation; O: oval; PC: pyriform to cylindrical; S: straight; OR: ovoidal rounded apex; T: basal truncation; C: clavate; R: reniform; BNP: barely notched and pointed; Cu: curved; NB: notched and blunt; PR: pedicellate and rounded; BN: barely notched; P Cu: papillate curved. ^b^ +: Present. ^c^ Mean value of 30 random selected conidia ± SD.

**Table 3 jof-09-00486-t003:** Sequence variation statistics of the partial sequences of three gene regions (ITS, LSU, and *tef1*) among the FSSC isolates examined in this study.

	No. of										Value for Indicated Neutrality Test ^b^			
Region, FSSC Species	Gene	Strains	Nucleotides	Haplotypes	Haplotype Frequency ^a^	Number of Segregating Sites	Polymorphic Sites (%)	Nucleotide Diversity (π)	Number of Mutation (ղ)	Haplotype (Gene) Diversity	Tajima’s *D*	Fu and Li’s *D**	Fu and Li’s *F**	Minimum No. of Recombination Events ^b^
*F. falciforme*	Concatenated	24	1556	12	0.50	24	1.542	0.00262	24	0.859	−1.47289 NS	−2.27745 NS	−2.37548 NS	0
	ITS	24	439	4	0.166	3	0.683	0.00181	3	0.634	−0.12171 NS	−0.18894 NS	−0.19630 NS	0
	*tef1*	24	668	10	0.416	18	2.694	0.00452	18	0.764	−1.46801 NS	−2.18497 NS	−2.29803 NS	0
	LSU	24	435	2	0.083	3	0.689	0.00059	3	0.083	−1.73253 NS	−2.52572* (*p* < 0.05)	−2.65835* (*p* < 0.05)	0
*F. keratoplasticum*	Concatenated	3	1556	2	0.666	1	0.064	0.00045	1	0.667	NA	NA	NA	0
	ITS	3	439	1	0.333	0	0.000	0.000	0	0.00	NA	NA	NA	0
	*tef1*	3	668	2	0.666	1	0.149	0.00105	1	0.667	NA	NA	NA	0
	LSU	3	435	1	0.333	0	0.000	0.000	0	0.000	NP	NP	NP	NP
FSSC 5	Concatenated	6	1556	6	1.000	10	0.642	0.00297	10	1.000	0.11945 NS	0.7918 NS	0.9479 NS	2
	ITS	6	439	2	0.333	1	0.227	0.00143	1	0.600	1.44510 NS	1.05247 NS	1.15768 NS	0
	*tef1*	6	668	6	1.000	9	1.347	0.00595	9	1.000	−0.11324 NS	−0.09221 NS	−0.10424 NS	1
	LSU	6	435	1	0.166	0	0.000	0.000	0	0.000	NP	NP	NP	NP
*F. vanettenii*	Concatenated	8	1556	4	0.50	5	0.321	0.00131	5	0.750	0.08445 NS	0.74709 NS	0.65390 NS	0
	ITS	8	439	3	0.375	4	0.911	0.00367	4	0.464	−0.01957 NS	0.56807 NS	0.47502 NS	0
	*Tef-1α*	8	668	2	0.25	1	0.149	0.00066	1	0.429	0.33350 NS	0.88779 NS	0.82528 NS	0
	LSU	8	435	1	0.125	0	0.000	0.000	0	0.000	NP	NP	NP	NP
Iran														
*F. falciforme*	Concatenated	17	1556	6	0.352	14	0.899	0.00218	14	0.721	−0.83607 NS	−1.90361 NS	−1.85065 NS	0
	ITS	17	439	2	0.117	1	0.227	0.00122	1	0.515	1.43020 NS	0.67700 NS	0.99442 NS	0
	*tef1*	17	668	6	0.352	10	1.497	0.00372	10	0.721	−0.73074 NS	−1.30215 NS	−1.68143 NS	0
	LSU	17	435	2	0.117	3	0.689	0.00083	3	0.118	−1.70573 NS	−2.25481 NS	−2.41419 NS	0
Non-Iran														
*F. falciforme*	Concatenated	Concatenated	7	1556	6	0.857	12	0.771	0.00313	12	0.952	−0.25712 NS	−0.07864 NS	−0.13028 NS
	ITS	ITS	7	439	3	0.428	3	0.683	0.00291	3	0.667	0.05031 NS	0.38925 NS	0.33832 NS
	*tef1*	7	668	5	0.714	9	1.347	0.00542	9	0.857	−0.35433 NS	−0.25556 NS	−0.30404 NS	0
	LSU	7	435	1	0.142	0	0.000	0.000	0	0.000	NP	NP	NP	NP

^a^ Number of haplotype/number of isolates. ^b^ NA: not applicable; NS: not significant; NP: not polymorphism.

## Data Availability

The data that support the findings of this study are available GenBank database (see Table 1).

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
