# Peer review of "Molecular Variability of the Fusarium solani Species Complex Associated with Fusarium Wilt of Melon in Iran"

_jof, 2023, doi:10.3390/jof9040486_

Round 1

Reviewer 1 Report

Please find the attachment for the incorporation of minor suggestions

Author Response

Dear Reviewer,

on behalf of all authors, we really appreciated your comments and suggestions and we accept all of them. we answered to your questions and addressed all the issues raised. Please see the attached file and note that the changes in the manuscript are also reported, for your convenience, in the response letter.

Kind regards

S. Olga Cacciola

Reviewer 2 Report

Dear authors,

The study is very informative. The data analysis is in accordance with the conclusions. The writing was easy to follow. In general, I would recommend the acceptance of the manuscript.

Author Response

Dear Reviewer,

on behalf of all authors, we really appreciated your positive comments about our manuscript.

Kind regards

S. Olga Cacciola

Reviewer 3 Report

This is a very well written manuscript that reports important new information regarding the Fusarium solani species complex (FSSC) and the genus Fusarium in the broad sense. Some specific comments relative to improving the presentation are below.

General comments: The authors might want to address more specifically any correlations that exist between the morphological characteristics presented in Table 2 and the lineages identified in phylogenetic analyses. Also, is there anything that can be said about the nature of human infections observed for members of some lineages (for example, the roles of risk factors such as being immunocompromised, age, comorbidity, etc) that would be of interest to readers who are not Fusarium experts?

Specific comments:

Line 118. I suggest either deleting “of” before “well-developed” or change wording.

Lines 162-163. I suggest changing “50 ul PCR” to “50 ul PCR reaction” in line 162 and “were used” to “was used” in line 163.

Footnote a, Table 2. Perhaps change “a truncate” be “a truncation” of change wording.

Figure 2 legend. Please add information describing the difference between the (a) and (b) panels.

Paragraph beginning line 285. Please reference Figure 3 (first indicated in line 293) earlier in this paragraph. Also, the reference to clade 3 (line 290) should be probably include a reference, as was done in earlier and later sections of the manuscript, and include some qualification, such as “previously recognized clade 3” or something to that effect.

Line 297. Perhaps “strongly-supported” rather than “strong-supported”

Table 3. Why are there separate columns for “Number of segregating sites” and “Number of mutations”? Also, some of the columns in the review manuscript are shifted out of position.

Line 422. I suggest changing “of FSSC population associated to” to “of the FSSC population associated with”

Line 439. I suggest changing “make more easily comparable” to “to make it easier to compare”

Line 440. Change “Authors” to “authors”

Line 441. Change “fall” to “fell” to be in agreement with “grouped”

Line 442. Perhaps change “the sequence of tef1 gene” to “sequences of tef1”. As it is writtne “sequence” does not agree with “were” in this sentence.

Line 457. Change “Consistently” to “Consistent” and “as causal;” to “as a causal”

Line 459. Perhaps change “encompasses also” to “also encompasses”

Line 476. Change the period after “this” to a comma.

Lines 515-516. The following phrase is confusing: “epidemiological implications and condition the disease management strategies”. How about changing the wording in this section to “have implications for epidemiology, disease management strategies, breeding programs for disease resistance, and quarantine measures”?

Author Response

(The authors gave the same response as above.)

Reviewer 4 Report

The present study is in my opinion well designed and well executed. I have some small comments.

The manuscript should be reviewed for small inconsistencies in English.

Along the text the authors refer to TCS, which should be written in its full for the first time it appears

Specific suggestions:

Line 27: One of the indicated keywords is N. keratoplastica. It should be written in its full form

Figure 2: Figure 2A should be remade because some bars and error bars seem to be truncated. If this is not the case, than the figure should be bigger to make it clearer. Moreover, in both 2A and 2B, the font size should be bigger to make it easier to read. In the legend of the figure, the authors must state what corresponds to 2A and 2B. Moreover, the authors should indicate how the value for the error bar was calculated.

Figure 3: Font size should be bigger, as some are difficult to read

Figure S1 could somehow be included in the manuscript as it may help better visualize the severity of different strains.

Author Response

(The authors gave the same response as above.)
